# Nanofiltration of Succinic Acid in Strong Alkaline Conditions

**DOI:** 10.3390/membranes9110147

**Published:** 2019-11-08

**Authors:** Klaus Schlackl, Richard Herchl, Wolfgang Samhaber

**Affiliations:** 1Kompetenzzentrum Holz GmbH, 4040 Linz, Austria; 2Lenzing AG, 4860 Lenzing, Austria; r.herchl@lenzing.com; 3Department of Process Engineering, Johannes Kepler University, 4040 Linz, Austria; wolfgang.samhaber@jku.at

**Keywords:** nanofiltration, NaOH, succinic acid, succinate, reflection coefficient

## Abstract

Nanofiltration is considered to be an appropriate separation technique in the production of bio-based materials. For the utilization of process streams from the viscose-fiber production, understanding the separation behavior of organic compounds in highly alkaline solutions is necessary. Experiments with succinic acid in sodium hydroxide (NaOH) solutions with varying concentrations up to 5 mol L^−1^ were performed with the NP030 membrane from Microdyn Nadir. Furthermore, experiments with aqueous disodium succinate and solutions of sodium sulfate in sodium hydroxide were carried out. The influence of concentration ratios and temperature was studied. The Spiegler and Kedem model as well as the Pusch model were applied to fit the experimental data. Additionally, scanning electron microscopy (SEM) and infrared (ATR–IR) measurements were performed to validate the chemical and thermomechanical stability of the membrane. The succinic acid retention varies with its degree of dissociation. In a fully dissociated form, the NaOH concentration shows no impact on the retention. In contrast, the retention of sulfate decreases with increasing NaOH concentration.

## 1. Introduction

Nanofiltration has the potential to be a highly effective separation method for the utilization of viscose-fiber production side streams. Furthermore, it can be used as a downstream process for the purification of process chemicals. A possible application is the removal of hemi celluloses and other wood degradation products from the highly caustic steeping lye. Hence, the lye could be recycled to the process and the demand of the make-up caustic solution can be significantly reduced. The total organic carbon (TOC) content can basically be divided into two fractions. The xylose oligo- and polymers with a molecular weight from ~600 to ~50,000 g mol^−1^ (beta fraction) and the hydroxyl acids, ranging from C_2_ up to C_6_ acids (gamma fraction).

The application of ceramic ultrafiltration membranes for the hemi cellulose removal (feed concentration: 30 g L^−1^) from steeping lye containing 18% sodium hydroxide was tested by Saurabh [1]. The used membranes had a molecular weight cut of (MWCO) of 3, 5, and 15 kDa. The membranes showed maximum retention of 0.68 (MWCO = 3 kDa), 0.67 (MWCO = 5 kDa) and less than 0.1 (MWCO = 15 kDa). Moreover, Saurabh [2] published results for polymeric ultrafiltration membranes for a similar feed stream. The retention was about 0.7 for the membrane with 3000 Da MWCO and decreased to about 0.15 (MWCO = 20,000).

Schlesinger [3] tested different polymeric nanofiltration membranes for an analog application. The membranes NTR-7470 (Nitto-Denko), MPF-34 (Koch), NP030 (Microdyn Nadir) and Osmonics GE (Desal) showed retention higher than 0.90 for molecules with a mass higher than 1000 g mol^−1^. This is significantly higher compared to the ceramic ultrafiltration membranes however there is still a low retention for the gamma fraction. A more detailed study on the retention of different components has not been performed yet. Furthermore intense fouling was observed by Schlesinger [4,5].

In addition to the viscose-fiber production, membrane processes in strong alkaline solutions are under investigation for streams of the cotton industry as studied by Son [6]. An additional application, which is under investigation, is the treatment of caustic cleaning agents used in the dairy industry [7,8,9]. It has to be mentioned that the sodium hydroxide concentrations in these applications are much smaller than in the viscose-fiber industry.

There is much research work that describes the transport of small organic and inorganic molecules through nanofiltration membranes. Different research groups [10,11,12,13] published the influence of the pH value, the zeta potential and the hydrophobicity of the membrane as well as the charge of the molecule on the transport mechanism. Furthermore, Simon [14] studied the impact of caustic cleaning on the membrane performance. All this research was done in pH ranges from 2 to 12, hence there is a lack of knowledge for higher pH values.

The influence of sodium hydroxide concentrations in the range of 1 up to 5 mol L^−1^ on the retention performance of nanofiltration membranes has not been studied so far. For several decades, highly alkaline stable nanofiltration membranes are available, although there are only few polymers available that can be used for such pH-robust membranes. One example is the membrane NP030 from Microdyn Nadir that is based on sulfonated polyethersulfone. Further, Advanced Membrane System Technologies supplies a highly alkaline stable nanofiltration membrane based on melamine polyamine.

Current research work is done to develop new membrane materials. Dalwani [15] developed a sulfonated polyether ether ketone (SPEEK)-based membrane with very high pH stability. Daems [16] developed a new technique to enhance the alkaline stability of PVDF membranes by grafting with polystyrene sulfonic acid.

Van Gestel [17] produced multilayer ceramic membranes with a MWCO lower than 200 Da. This was achieved with α-Al_2_O_3_/γ-Al_2_O_3_/anatase membranes that are stable in the pH range from 3 to 11. At higher and lower pH values, the combination anatase on an α-Al_2_O_3_ performs better.

In the present work, succinic acid was chosen as the model compound to investigate the retention behavior of gamma components due to several reasons. It has a molecular weight of 118 g mol^−1^, which is in the middle between acetic acid (60 g mol^−1^), and glucoisosaccharinic acid (180 g mol^−1^), two representative gamma components. Furthermore, its anion can act as mono- or divalent ion depending on the pH value.

A better understanding of the NaOH concentration impact on the retention behavior is important for the design of an efficient nanofiltration process, as partial dilution can improve the caustic recovery rate. The comparison of disodium succinate and sodium sulfate should help to improve the knowledge of membrane retention mechanisms and could indicate the possibility of separation between organic and inorganic divalent ions.

## 2. Material and Methods

### 2.1. Material

Succinic acid with a purity of 99.9% and disodium succinate with a purity of 99% were obtained from Sigma Aldrich. Hydroxide solution was prepared from deionized water and dry pellets with a purity of 99% (Sigma Aldrich, Taufkirchen, Germany).

NP030 membrane samples from Microdyn Nadir (Wiesbaden, Germany) in DIN A4 format were purchased directly from the manufacturer and were used without any pretreatment. Membrane properties are shown in Table 1.

### 2.2. Nanofiltration

Experiments were performed on two different lab-scale nanofiltration rigs. One was a modified OSMO-MC-01 flat-sheet lab-scale device from OSMOTA (Rutesheim, Germany), the other one was a MEMCELL 3 from OSMO (Korntal-Münchingen, Germany). A heat exchanger was added in the feed stream of both systems for temperature regulation. The membrane area was 80 cm^2^; for the OSMO-MC-01 and three times 80 cm^2^ in the MEMCELL 3 device. The cells were arranged in parallel. Permeate and retentate were returned to the feed tank to keep the feed concentration constant. The samples amount was kept low (~1 mL) compared to the feed volume (5 L). The feed concentration was measured throughout the experiment to ensure stable concentrations.

The refractive index of permeate was observed to identify if a quasi-stationary state was achieved. Samples of at least six different pressures were collected with each combination of parameters.

All experiments at a certain NaOH concentration were performed with the same membrane. Before the experiment, the membrane was rinsed with a NaOH solution until constant permeate flux (approx. 50–80 h). The NaOH concentration during this pretreatment was equal to the concentration in the following experiment.

### 2.3. Refractometer

Refractive index measurements were performed with the refractometer Abbemat 500 from Anton Paar (Graz, Austria). The refractometer was used on the one hand to check constant permeate composition and on the other hand to monitor constant feed composition.

Furthermore, the refractive index measurements were used to measure the NaOH concentration.

Calibration data were measured for different solutions of succinic acid (0–0.85 mol L^−1^) in different NaOH concentrations (0–5 mol L^−1^). The correlation of the refractive index (*n*) based on NaOH concentration (*c_NaOH_* in mol L^−1^) and succinic acid concentration (*c_SA_* in mol L^−1^) is given by Equation (1).
(1)$$n=A+B\;\times \;c_{NaOH}+C\;\times \;c_{SA}+D\;\times \;c_{NaOH}\;\times \;c_{SA}+E\;\times \;c_{NaOH}^{2}+\;F\;\times \;c_{SA}^{2}$$
where *A* = 1.333350; *B* = 0.010348; *C* = 0.008236; *D* = −0.001227; *E* = −0.000337; *F* = −0.000284.

Based on this correlation, the NaOH concentration can be calculated for a given succinic acid concentration.

### 2.4. Titration

To verify the calculated NaOH concentration (based on the refractive index), a batch of samples was additionally analyzed by titration with 0.25 mol L^−1^ sulfuric acid to pH 3.7.

To eliminate the influence of the titrant, preliminary tests were done with sulfuric acid and hydrochloric acid. No significant influence of the titrant was observed.

Furthermore, preliminary tests showed that succinic acid is completely undissociated at pH 3.7. This needs to be considered for the calculation of the sodium retention based on titration.

### 2.5. Chromatography

Succinic acid and disodium succinate were quantified by high performance liquid chromatography (HPLC) analysis. A Thermo Hypersil GOLD aQ 250 mm column with the photodiode array detector Dionex UVD 340U was used. A wavelength of 210 nm was applied for detection. Sulfate concentration was analyzed using ion chromatography. A CarboPac PA10 was used as a column. The detector for quantification was a Dionex ED 50. All the HPLC instruments used were purchased from Thermo Fisher Scientific (Vienna, Austria). The data processing was done with the software Chromeleon (version 7.2).

### 2.6. Scanning Electron Microscopy

SEM measurements were done for different membranes before and after the experiments. The used membrane samples were washed with deionized water and subsequently dried. The unused membrane sample was used as received from the manufacturer.

The samples were gold-plated prior to the analysis. Images were taken using a FEI Quanta 450 with 10 Kv Thermo Fisher Scientific (Vienna, Austria). For the image processing the software Phenom ProSuite (version 2.9.0.0) was used.

### 2.7. Attenuated Total Reflectance Infrared Spectroscopy

All analyzed membrane samples except the unused one were washed with deionized water for one minute after exposure to NaOH solution.

Measurements were done with a Bruker Tensor 27 Spectrometer, which was equipped with a Specac Single Reflection Diamond ATR “Golden Gate” (Bruker Austria GmbH, Vienna, Austria). The software OPUS (version 7.5). was used for data processing.

## 3. Theoretical Background

Two different models were used to assess experimental data.

### 3.1. Spiegler and Kedem Model

The Spiegler and Kedem model [21] is based on the thermodynamics of irreversible processes. The flux of the solvent *J_L_* is defined by:(2)$$J_{L}=-P_{L}\;\times \;\left(\frac{dp}{dz}-\sigma \;\times \;\frac{d\Pi }{dz}\right)$$
where *P_L_* is the permeability of solvent, *dp*/*dz* the pressure gradient through the membrane and *dπ*/*dz* the osmotic pressure gradient. *σ* represents the reflection coefficient.

The flux of the solute *J_S_* is given by:(3)$$J_{S}=-P_{S}\;\times \;\frac{dc_{S}}{dz}+\left(1-\sigma \right)\;\times \;c_{S}\;\times \;J_{L}$$

*P_S_* is the permeability of the solute, *dc_S_*/*dz* the concentration gradient and *c_S_* the concentration of solute. The first term is related to transport due to diffusion and the second one represents convective transport. If *σ* = 1 the Spiegler and Kedem model conforms to the solution–diffusion model. With the common definition of the retention *R*, as expressed in Equation (4),
(4)$$R=1-\frac{c_{Perm}}{c_{Ret}}$$
and further calculations, the retention can be written as a function of the overall permeate flux *J_V_* and the two model coefficients *σ* and *P*:(5)$$R=1-\frac{1-\sigma }{1-\sigma \;\times \;\exp\left(\frac{\left(\sigma -1\right)\;\times \;J_{V}}{P}\right)}$$
with:(6)$$P=\frac{P_{S}}{z}$$

By fitting the experimental data, *σ* and *P* can be evaluated for any combination of temperature and concentrations.

### 3.2. Pusch Model

The Pusch model is another well-known model, which can also be used to describe the nanofiltration processes. It is based on thermodynamics of irreversible processes and was expounded by Pusch in 1977 [22]. The equation describes the reciprocal value of the retention (*R*) as a function of the reciprocal retention at infinite flux (*R_∞_*), two constants (*L_D_*, *L_P_*), the osmotic pressure difference (π) and the reciprocal value of the flux (*J_V_*).
(7)$$\frac{1}{R}=\frac{1}{R_{\infty }}+\left(\frac{L_{D}}{L_{P}}-{R_{\infty }}^{2}\right)\;\times \;\frac{L_{P}\;\times \;\Pi_{1}}{R_{\infty }\;\times \;J_{V}}$$

To calculate *R_∞_* the reciprocal value of the flux is plotted on the x-axis and the reciprocal value of the retention on the y-axis. A linear correlation of this set of data points gives the reciprocal values of *R_∞_* as the ordinate intercept.

### 3.3. Law of Electroneutrality

Due to the law of electroneutrality, the number of negatively charged groups must be equal to the number of positively charged groups. Succinic acid completely dissociates in highly alkaline solutions. Consequently, the succinate has two negatively charged groups per molecule. Due to the absence of other cations, the sodium ions equalize the negative charge of the succinate. Because of this, the retention of sodium is related to the retention of the succinate as shown here:(8)$$c_{R,Na}\;\times \;R_{Na}=c_{R,Su}\;\times \;R_{Su}\;\times \;2$$

*c_R,Na_* and *c_R,Su_* represent the concentration of sodium and succinate in the retentate. *R_Na_* denotes the retention of sodium and *R_Su_* indicates the retention of succinate. Equation (8) applies only to strongly alkaline solutions in which the amount of hydronium ions is negligible and thus sodium is the only cation in the solution.

## 4. Results and Discussion

### 4.1. Saturation Solubility of Succinic Acid

The saturation solubility of succinic acid increases with increasing concentration of NaOH as shown in Figure 1. The solubility in 5 M NaOH is four times higher than in water. Succinic acid in protonated form is less soluble than the salt form. An increasing NaOH concentration enhances the dissociation and therefore the solubility. Solubility at 40 °C is on average 0.24 mol L^−1^ higher than at 30 °C. The solubility of disodium succinate in water is approximately twice as high as that of succinic acid. Thus, the succinic acid and disodium succinate concentrations used in the nanofiltration experiments were far below the solubility limits.

### 4.2. Flux

Figure 2 shows the permeate flux of water and NaOH solutions with different concentrations. As can be seen, the flux is linearly dependent on the applied pressure in the range of 5 to 30 bar.

Figure 3 shows the influence of the NaOH concentration on the permeability coefficient (*L_P_*) at 40 °C. By increasing the NaOH concentration from 0 to 5 mol, L^−1^
*L_P_* decreases from 2.05 to 0.15 kg m^−2^ h^−1^ bar^−1^. The flux decrease by increasing NaOH concentration occurs for two different reasons. Firstly, the viscosity increases from 0.65 mPa·s (water) to 2.5 mPa·s (5 M NaOH). The second reason is the difference in osmotic pressure. The NP030 shows a small retention for NaOH. Consequently, an osmotic pressure difference occurs leading to a reduction of permeate flux. The higher the feed concentration, the higher is the osmotic pressure difference.

### 4.3. Retention of Sodium Sulfate

The retention value of sulfate, according to manufacturer data, is between 0.80 and 0.95. The experiments were performed with 40 bar at 20 °C, details on the feed concentration were not given. Sulfate retention was studied to compare the influence of the structure of two different divalent ions. The organic succinate ion has an oval structure with the negatively charged groups on both ends. In contrast, the sulfate ion is circular. Figure 4 shows the retention of sulfate with a feed concentration of 0.1 mol L^−1^ sodium sulfate. As can be seen in the performed experiments, the retention of sulfate in aqueous solution was lower than 0.8 for permeate fluxes lower than 30 kg m^−^^2^ h^−1^. At higher fluxes, the sulfate retention was in the range specified by the supplier. The reflection coefficient found in the Spiegler and Kedem model was calculated to be 0.9.

Furthermore, it can be seen that the sulfate retention is reduced by adding NaOH to the solution. The higher the NaOH concentration is, the lower the sulfate retention.

### 4.4. Retention of Succinic Acid

Figure 5 shows the retention at infinite permeate flux calculated with the Pusch model (*R_∞_*) in dependency of the temperature, the feed concentration of succinic acid and the NaOH concentration. Figure 6 shows the reflection coefficient (*σ_SA_*) of the Spiegler and Kedem model with the same parameters. The standard deviation for the Spiegler and Kedem fit is significantly smaller than for the Pusch Model. Consequently, the *R_∞_* values show a higher variance than the *σ_SA_* values. Hence, further calculations were performed only with the Spiegler and Kedem model.

As it is shown in Figure 6, the reflection coefficient of succinic acid is independent from the NaOH concentration as long as the succinate ion is completely dissociated. The incompletely dissociated form shows significantly lower *σ_SA_* values. The *σ_SA_* values at 0.43 mol L^−1^ succinic acid in 1 molar NaOH solution are lower compared to the data points at complete dissociation. 

The experiments with 0.2 mol L^−1^ succinic acid were performed twice, once as a first experiment of a series and a second time at the end of a series. As shown, the results have a good reproducibility.

The succinate ion shows a similar retention in 0.1 molar aqueous disodium succinate solution (chapter 4.6.) as sulfate in a 0.1 molar sodium sulfate solution. The situation changes for the sulfate ion in a caustic solution. Its retention decreases with increasing NaOH concentration. In comparison, the succinate retention is unaffected from NaOH in a wide concentration range. A difference in the capability of hydration shell formation can explain this. It seems that the succinate ion is able to pull the water molecules stronger than the sulfate ion. Bouchoux [23] pointed out, that the sodium lactate retention is almost unaffected by adding sodium chloride to the solution, in contrast glucose retention decreased significantly. This means organic anions like lactate and succinate are able to form hydration shells, which are less affected from competitive hydration of other ions in the solution. Hence, the type and concentration of co-ions in the solution can control the separation performance of ions. To prove this assumption, further investigations with molecules with different structures are necessary.

In the case of incomplete dissociation of the succinic acid, the retention decreases by about half. This is within the same range as the chloride retention specified by the manufacturer (0.25–0.35).

### 4.5. Retention of Sodium Ions

Figure 7 shows the reflection coefficient of sodium ions (*σ_Na_*). As mentioned in Equation (8) the retention of sodium ions is related to the retention of succinate ions by the law of electroneutrality, assuming that no further cations are present. In solutions containing excess NaOH with respect to the amount of carboxylic acid groups, this requirement is met. According to Equation (8) the slope of the data points for 1 M NaOH is five times steeper than for 5 M NaOH. The calculated slopes are 1.09 (1 M NaOH) and 0.22 (5 M NaOH). The decrease of the pH value leads to the presence of hydronium ions, which decouples the retention between the sodium and succinate ions. Thus, this point was not taken into account for the slope calculation.

Additionally, the retention of sodium ions was calculated with Equation (8). The results are given as crosses in Figure 7. As can be seen, the experimental data and the calculated values are in good agreement.

### 4.6. Retention of Disodium Succinate

To investigate the retention of equimolar solutions of sodium ions and carboxylic groups, experiments with disodium succinate were performed. Figure 8 shows the reflection coefficient in relation to feed concentrations. As can be seen, the retention for disodium succinate in water is higher than the retention of succinic acid in NaOH solutions. Furthermore, it can be seen, that the reflection coefficient decreases with increasing feed concentration.

### 4.7. SEM Measurements

Rinsing the membrane with water leads to a tripling of the active membrane layer thickness due to a swelling effect. Swelling in caustic solution leads to the same result (Figure 9).

No changes of the active layer thickness can be seen by samples of the used membranes stressed up to 25 bar at 40 °C. The shape of the pores in the first support layer changes from an oval shape to an elongated shape. This does not influence the overall membrane performance.

By increasing the pressure up to 55 bar (40 °C), a significant change can be observed, see Figure 10. The small pores of the first support layer disappear and a compact layer is formed. The compression of this support layer is irreversible and leads to a significantly lower permeate flux. The bigger pores in the bottom part of the first support layer are still present.

### 4.8. ATR–IR Measurement

Figure 11 shows ATR-IR measurements of different conditioned membranes. 

Essentially, the spectra of all tested membranes resemble the spectrum of PES [Poly(p-phenylene ether sulfone)] which is declared as the main membrane polymer by the supplier.

The band at a wavelength from 3200–3500 cm^−1^ represents the O–H stretching band. In contact with NaOH solution, a change in the ATR-IR diagram can be observed but is not related to any changes in the membrane polymer composition. These changes appear within the first hour and do not alter significantly over time.

The bands at 2934 and 2879 cm^−1^ represent aliphatic C-H stretching bonds which decrease in contact with NaOH solution. The bands at 1040 and 925 cm^−1^ refer to glycerol, which was used for membrane preparation or storage and is washed out immediately by the solutions. The band at 1400 cm^−1^ indicates decompositions of carbonates on the membrane. Carbonates were formed from carbon dioxide, which is dissolved in the solution. By longer exposure to the NaOH solution, the band increases slightly. None of the characteristic polymer bands changes significantly with the treatment time. Consequently, NP030 can be considered as chemically stable in alkaline solutions with concentrations up to 5 mol L^−1^ NaOH. This is in good agreement with the data published by Schlesinger [3], which indicates high alkali resistance of the NP030.

## 5. Conclusions

Based on the retention results of aqueous solutions, it can be concluded that the organic salt (disodium succinate) behaves similarly to the inorganic sodium sulfate. A different retention behavior is observed in NaOH solutions. Sulfate ions are more affected by the presence of co-ions than succinate ions. Degree of dissociation is the factor with the highest impact on the retention of succinic acid. Consequently, the charge selectivity of the NP030 nanofiltration membrane could be verified with an organic salt by changing its degree of dissociation. A higher NaOH concentration in the feed solution leads to a significant decrease of permeate flux.

The retention of completely dissociated succinate, is unaffected by the NaOH concentration. Hence, a dilution during the nanofiltration process may improve the process efficiency, as the flux increases. Furthermore, the selectivity between sulfate and succinate can be controlled by the NaOH concentration.

The results of the ATR–IR measurements and the thermomechanical stress tests indicate an applicability of the NP030 membrane in highly alkaline conditions. Further performance tests with model solutions are necessary to prove this.

Furthermore, it is shown, that the Spiegler and Kedem model appropriately describes the retention behavior of all investigated species, even in five molar NaOH.

## Figures and Tables

**Figure 1 membranes-09-00147-f001:**
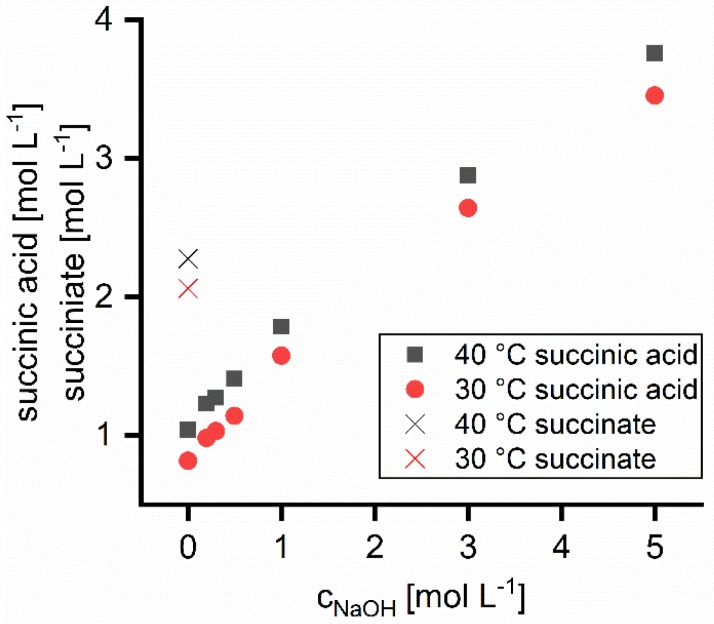
Saturation concentration of succinic acid in NaOH solutions at 30 and 40 °C (dots). Crosses mark the saturation solubility of disodium succinate in water.

**Figure 2 membranes-09-00147-f002:**
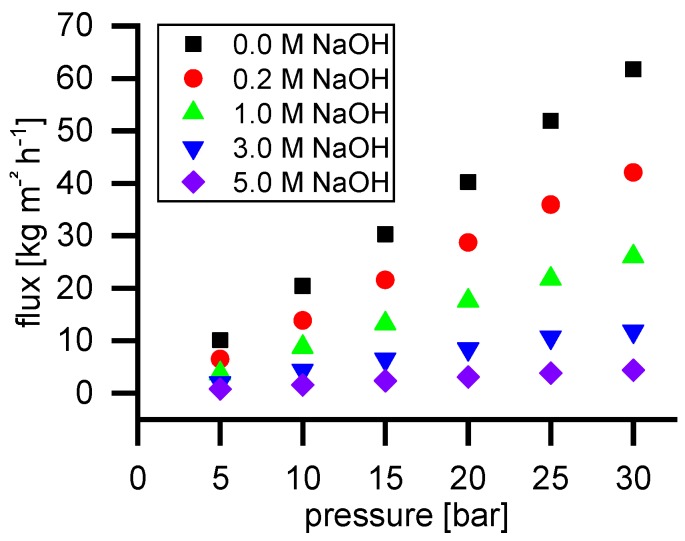
Permeate flux of water and NaOH solutions with different concentrations in the range from 5 to 30 bar at 40 °C.

**Figure 3 membranes-09-00147-f003:**
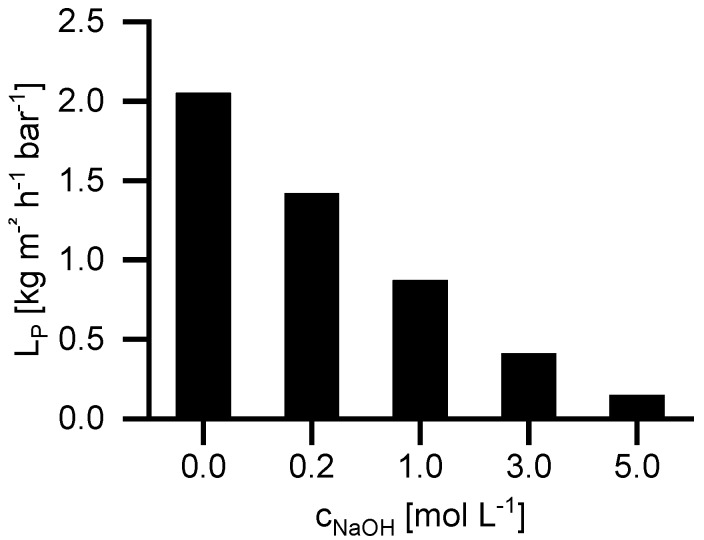
Permeability coefficient (*L_P_*) of water and NaOH solutions of different concentrations at 40 °C.

**Figure 4 membranes-09-00147-f004:**
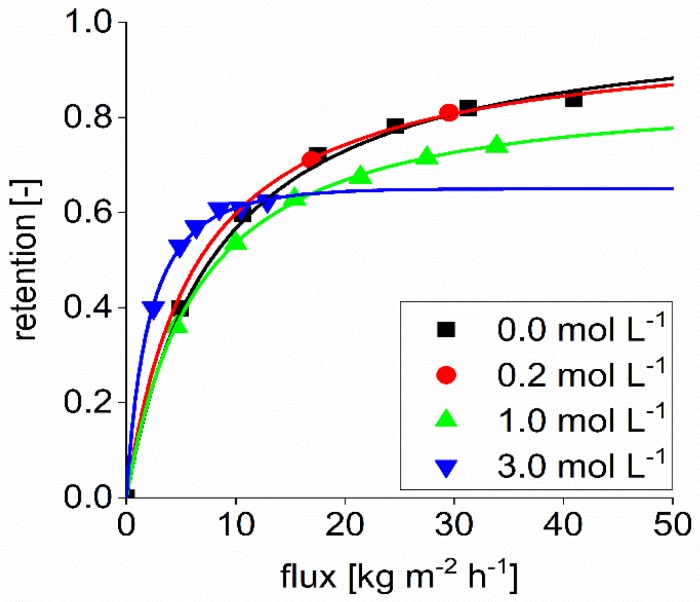
Retention of sulfate in pure water and in mixtures of different NaOH concentrations at 40 °C. The dots indicate experimental data, the solid line shows the Spiegler and Kedem fit.

**Figure 5 membranes-09-00147-f005:**
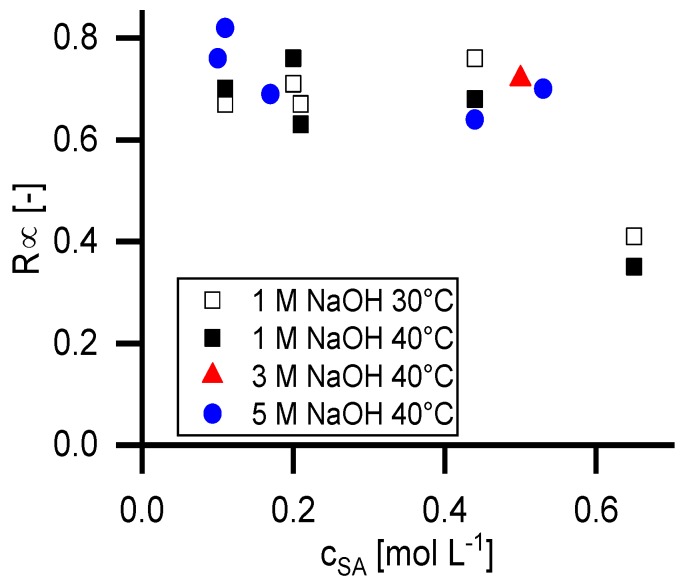
Retention of succinic acid at infinite permeate flux (*R_∞_*) in dependency of feed concentration calculated with the Pusch model. NaOH concentration and temperature are shown as process parameters for the generated data.

**Figure 6 membranes-09-00147-f006:**
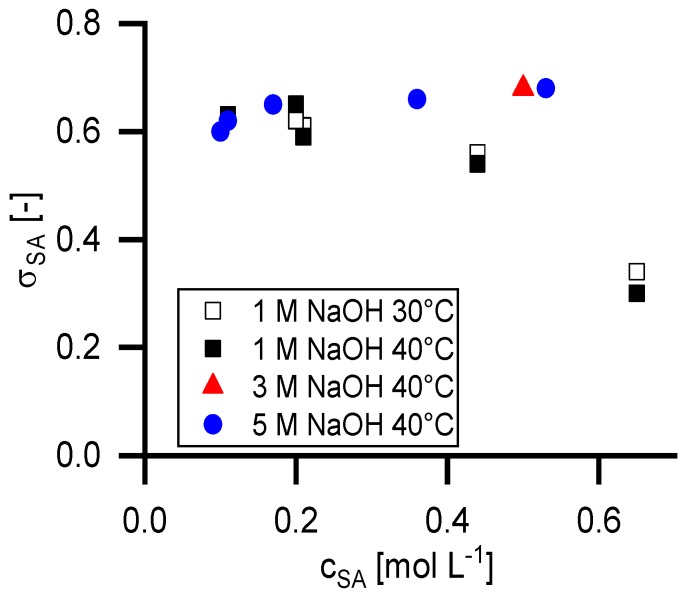
Reflection coefficients of succinic acid in dependency of feed concentration according the Spiegler and Kedem model.

**Figure 7 membranes-09-00147-f007:**
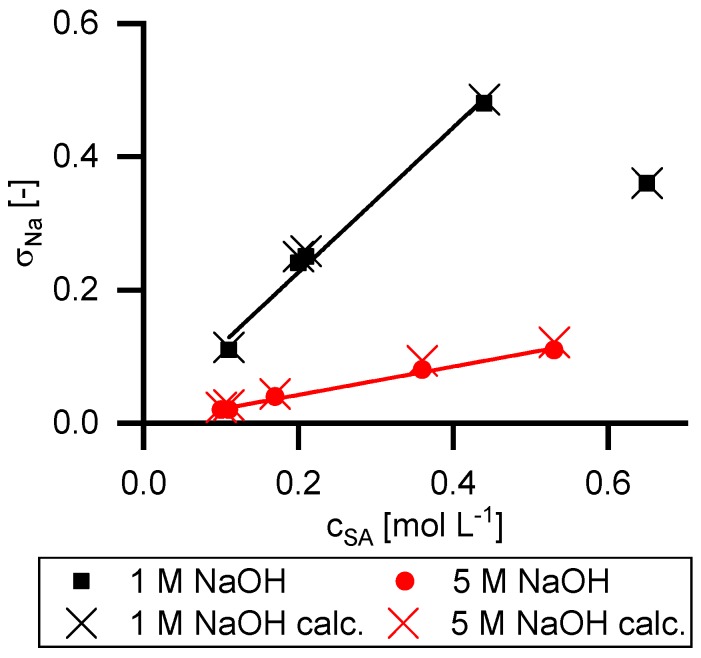
Reflection coefficients of the sodium ions in dependency of succinic acid feed concentration. The dots represent calculated values with the Spiegler and Kedem model, based on experimental data. Crosses represent the calculated reflection coefficients according to Equation (8).

**Figure 8 membranes-09-00147-f008:**
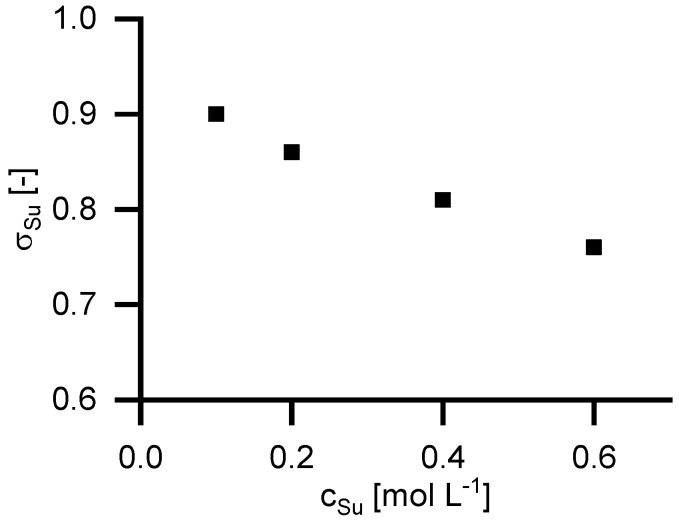
Reflection coefficient of disodium succinate in aqueous solution at 40 °C in relation to the feed concentration of succinate.

**Figure 9 membranes-09-00147-f009:**
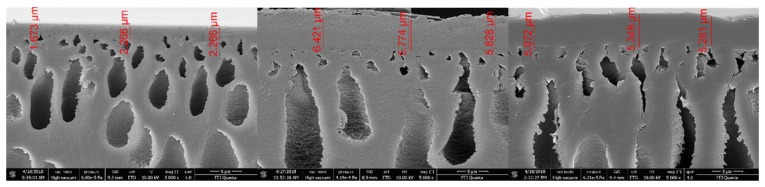
SEM measurements of a new membrane (**left**), a pre-conditioned membrane (**middle**), and a used membrane at 40 °C and 25 bar (**right**). The shown values represent the active membrane layer thickness.

**Figure 10 membranes-09-00147-f010:**
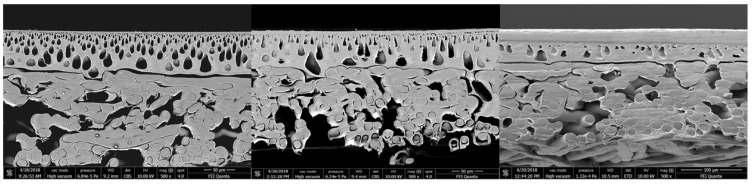
SEM measurement of the whole cross-sections of an unused membrane (**left**), a lowly stressed (max. 25 bar) membrane (**middle**), and a highly stressed (max. 55 bar membrane) (**right**).

**Figure 11 membranes-09-00147-f011:**
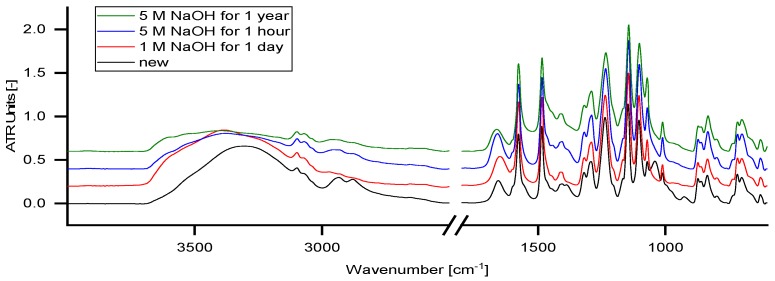
ATR–IR spectra of membranes stored in NaOH solutions with different concentrations (1–5 molar) for varying periods (1 h–1 year).

**Table 1 membranes-09-00147-t001:** Overview of the membrane specifications found in the literature and on the material data sheet provided from the manufacturer.

Parameter	Value	Source
Na_2_SO_4_ retention ^1^ [-]	0.80–0.95	Microdyn Nadir
MWCO (Da)	520	Kovac [18]
MWCO (Da)	400	Boussu [19]
Water permeance [L m^−2^ h^−1^ bar^−1^]	3.8	Boussu [19]
Water permeance [L m^−2^ h^−1^ bar^−1^]	4.4	Bargeman [20]
pH stability	1–14	Microdyn Nadir
Zeta potential pH 3 (mV)	1	Boussu [19]
Zeta potential pH 7 (mV)	−15	Boussu [19]
Zeta potential pH 12 (mV)	−20	Boussu [19]
Contac angle (°)	88	Boussu [19]
Membrane material	permanently hydrophilic polyethersulfone	Microdyn Nadir

^1^ Test conditions: 40 bar, 20 °C, stirred cell (700 rpm).

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
