# Peer review of "Nanofiltration of Succinic Acid in Strong Alkaline Conditions"

_membranes, 2019, doi:10.3390/membranes9110147_

Round 1

Reviewer 1 Report

The manuscript tested the performance of nanofiltration for succinic acid separation from strong alkaline condition. According to their results, the selected NF membrane can be used in such conditions. However, the current form of the manuscript needs major improvement before being considered for publication in Membranes. Below are some questions/concerns that I hope the authors could address.

The author measured the separation performance of the NF membrane to sodium sulfate, but didn’t give the reason. It has no influent to the separation of succinic acid. Table 2 can be deleted. To study the stability of the NF membrane in NaOH, the separation performance of succinic acid should be tested. Only ATR-IR is not enough. In Fig. 4, only 2 or 3 point for the 1 M and 3 M result cannot be compared to the model and not enough to calculate the reflection coefficient.

Author Response

Dear reviewer,

Thank you for your clear and helpful comments on my manuscript. I focused on your comments and did major revisions. A detailed list of the changes is shown here:

The relevance of the sulfate retention measurements were added to the introduction (lines 82-87) and the results chapter (lines 228-230). Explanations were added to the introduction (lines 82-87) and to the conclusion (lines 379-382) to illustrate the relevance of the publication in more detail. Table 2 (line 325) was deleted, as the relevant data are shown in Figure 11. The conclusions based on the ATR-IR measurements were adapted in the lines 383-385. Additional experiments were performed to add more data to figure 4 (line 237) to enable a more accurate fit.

Based on the comments of reviewer 2, a single “Results and discussion” chapter was written!

All the changes are marked with a notice in which the number of the reviewer is announced. Changes based solely on merging the results and the discussion chapter have not been tagged with the reviewer's number to keep it clear.

Reviewer 2 Report

The paper presents an experimental study of the retention of succinic acid from highly alkaline solutions using a commercial negatively charged nanofiltration membrane. The experiments were conducted carefully, and the data are sound. The discussion needs major strengthening. Please see below more specific comments:
1. Equation 8: the hydronium cation and the hydroxyl anion also come into play here and chapter 3.3 is not exactly accurate; changes in pH can occur that unbalance this equation
2. Lines 228-232: Figure 6 presents the reflection coefficient, not the retention; therefore, this discussion may be confusing. The two parameters are related but they are not the same thing.
3. I do not think that authors can draw a conclusion about differences in retention at the two temperatures (line 232), based on the data.
4. Authors show a large amount of data without adequate discussion. The Discussion chapter is underwhelming. I suggest discussing the data more thoroughly as they are presented, in a unique Results and Discussion chapter.
5. What is missing is a continued discussion on the implications of the results and of the retention mechanisms on real application.

Author Response

Dear reviewer,

Thank you for your clear and helpful comments on my manuscript. I focused on your comments and did major revisions. A detailed list of the changes is shown here:

1.       A more precise formulation for Eq. 8 is given in the line 188 and the lines 193 – 195 and 283 - 285.

2.       The expression retention was replaced by the correct terms “reflection coefficient” OR “σSA”. (lines 249-253)

3.       The statement has been deleted as the accuracy of the data is not considered sufficient to draw this conclusion. (line 253)

4.       The discussion chapter was strengthened and one single Results and discussion chapter was written. (lines 196 – 370)

5.       Explanations were added to the introduction (lines 82-87) and to the conclusion (lines 379-382) to illustrate the relevance of the study in more detail.

All the changes are marked with a notice in which the number of the reviewer is announced. Changes based solely on merging the results and the discussion chapter have not been tagged with the reviewer's number to keep it clear.

Round 2

Reviewer 1 Report

The manuscript can be accepted as it is.